# Implementation costs and cost-effectiveness of ultraportable chest X-ray with artificial intelligence in active case finding for tuberculosis in Nigeria

Tushar Garg[1], Stephen John[2,3], Suraj Abdulkarim[2,3], Adamu D. Ahmed[2], Beatrice Kirubi[1], Md. Toufiq Rahman[1], Emperor Ubochioma[4], Jacob Creswell[1]*

**1** Stop TB Partnership, Geneva, Switzerland, **2** Janna Health Foundation, Yola, Adamawa State, Nigeria, **3** SUFABEL Community Development Initiative, Gombe, Gombe State, Nigeria, **4** National TB and Leprosy Program, Federal Ministry of Health Nigeria, Abuja, Nigeria

* jacobc@stoptb.org

## Abstract

Availability of ultraportable chest x-ray (CXR) and advancements in artificial intelligence (AI)-enabled CXR interpretation are promising developments in tuberculosis (TB) active case finding (ACF) but costing and cost-effectiveness analyses are limited. We provide implementation cost and cost-effectiveness estimates of different screening algorithms using symptoms, CXR and AI in Nigeria. People 15 years and older were screened for TB symptoms and offered a CXR with AI-enabled interpretation using qXR v3 (Qure.ai) at lung health camps. Sputum samples were tested on Xpert MTB/RIF for individuals reporting symptoms or with qXR abnormality scores ≥0.30. We conducted a retrospective costing using a combination of top-down and bottom-up approaches while utilizing itemized expense data from a health system perspective. We estimated costs in five screening scenarios: abnormality score ≥0.30 and ≥0.50; cough ≥ 2 weeks; any symptom; abnormality score ≥0.30 or any symptom. We calculated total implementation costs, cost per bacteriologically-confirmed case detected, and assessed cost-effectiveness using incremental cost-effectiveness ratio (ICER) as additional cost per additional case. Overall, 3205 people with presumptive TB were identified, 1021 were tested, and 85 people with bacteriologically-confirmed TB were detected. *Abnormality ≥ 0.30 or any symptom* (US$65704) had the highest costs while *cough ≥ 2 weeks* was the lowest (US$40740). The cost per case was US$1198 for *cough ≥ 2 weeks*, and lowest for *any symptom* (US$635). Compared to baseline strategy of cough ≥ 2 weeks, the ICER for *any symptom* was US$191 per additional case detected and US$ 2096 for *Abnormality ≥0.30 OR any symptom* algorithm. Using CXR and AI had lower cost per case detected than any symptom screening criteria when asymptomatic TB was higher than 30% of all bacteriologically-confirmed TB detected. Compared to traditional symptom screening,

**Data availability statement:** All the data necessary to reproduce the cost-effectiveness analysis is available within the manuscript, and the code for this analysis is provided on GitHub at https://github.com/peptonefizz/Costing-Nigeria-Analysis. The case-finding data is available in a previous paper [26].

**Funding:** This study was funded by the Stop TB Partnership's TB REACH Initiative, through funding from Global Affairs Canada grant number CA-3-D000920001 (https://w05.international.gc.ca/projectbrowser-banqueprojets/project-projet/details/D000920001). The funder had no role in study design, data collection and analysis, decision to publish, or preparation of the manuscript. TG, BK, MTR, and JC work at the Stop TB Partnership and are funded in part by TB REACH.

**Competing interests:** TG, BK, MTR, and JC work at the Stop TB Partnership which supplies portable CXR equipment and AI software through its Global Drug Facility. None of the authors are involved in any procurement or supply chain work.

using CXR and AI in combination with symptoms detects more cases at lower cost per case detected and is cost-effective. TB programs should explore adoption of CXR and AI for screening in ACF.

## Author summary

In our study, we explored strategies to enhance tuberculosis (TB) screening in active case finding (ACF) campaigns using ultraportable chest X-ray (CXR) machines with artificial intelligence (AI) in remote areas of Northeast Nigeria. We organized health camps where individuals over 15 years were screened for TB symptoms and given CXR with AI-enabled interpretation to detect TB abnormalities. If someone showed symptoms or their CXR AI score suggested TB, we confirmed with sputum testing using Xpert MTB/RIF. Our analysis compared different screening methods, examining the costs, number of TB cases detected, and cost-effectiveness. The combined approach of symptom screening plus CXR with AI proved to be more effective, less expensive than screening for only cough, and cost-effective for the setting. Key cost drivers were community mobilization, CXR equipment and AI license, and GeneXpert equipment. Using CXR and AI had lower cost per case detected than any symptom screening criteria when asymptomatic TB was higher than 30% of all bacteriologically-confirmed TB detected. Our findings suggest that TB programs could consider using CXR and AI to improve case detection in similar settings, making TB screening both more accessible and cost-effective.

## Background

Chest radiography (CXR) has long played a critical, and sometimes maligned, role in screening and diagnosis of tuberculosis (TB). Since the turn of the century, and based on a better understanding of the importance to identifying all people with TB, not just those who have advanced disease [1], recommendations around active case finding [2], findings from modern TB prevalence surveys demonstrating the value of CXR to detect asymptomatic TB [3,4], the scale-up of more sensitive and expensive molecular diagnostics [5], and most recently the first-of-its-kind WHO recommendation to use artificial intelligence (AI) to interpret CXR images [2] coupled with the introduction of ultraportable X-ray equipment [6], the TB community has embraced outreach efforts to identify people with TB [7,8], increasingly with CXR as an integral component [9–12].

The benefits of ACF have been well documented [13,14], and numerous studies have reported on the costs and cost effectiveness of outreach efforts [15–17]. It is clearly more costly to mobilize teams outside of health facilities to screen people for TB than passive approaches. Nevertheless. ACF reaches more people, earlier in their disease progression and can reduce patient costs [18]. It is also well established that placing CXR in diagnostic algorithms as a screening tool, can improve the

performance of the algorithms compared to symptom screening [19,20], but despite advances, x-ray technology is also an expensive tool which adds to the costs of any ACF approach.

Despite the growing interest in CXR screening and AI technology, there exists scant evidence of the costs of using CXR in ACF. A randomized trial from Malawi showed that chest X-ray screening with computer-aided interpretation for TB among symptomatic individuals attending a primary health center with universal HIV screening reduced time to treatment and improved yield of HIV and TB detection, but was not cost effective within the eight week trial analysis [21]. A modeling study from Pakistan estimated the costs of country-wide ACF with CXR comparing human reading and computer aided interpretation as well as across deployment modes of the AI concluding that the per image cost at scale would be cheaper with AI than human readers and that unlimited reads would be less expensive than pay per image charges [22]. An early evaluation of AI for CXR modeled data from the facility based TB-NEAT study [23] to calculate diagnostic costs of using CXR and AI to triage people for Xpert testing and found it improve the daily screening throughput and substantially reduce testing costs [24]. Finally, a study from Brazil in three prisons compared four different screening algorithms and found that using CXR with an early version of CAD4TB (v5, Delft Imaging) with a threshold score of 60 was more costly and less sensitive than testing everyone with Xpert who could produce sputum regardless of symptoms [25]. However, we could not identify, any study comparing costs of ACF with and without CXR. As national TB responses prepare to deploy new technology to improve TB diagnosis, especially in key populations, more data on the costs, and cost drivers of such interventions are needed.

In this costing study, we build on the results of an ACF intervention in Northeast Nigeria using ultraportable chest X-ray and AI for screening with molecular diagnostic testing using Xpert MTB/RIF (Xpert). We aimed to estimate the costs associated with implementing screening camps utilizing ultraportable CXR combined with AI software within the framework of ACF and compare how different screening algorithms would perform.

## Methods

### Study design, population, setting

We provide a brief summary of the ACF intervention for TB in Northeast Nigeria; details published previously [26]. We conducted a series of community camps in 15 local government areas (LGA, similar to sub-districts) in two states, Gombe and Adamawa, that were purposefully selected to target services to nomadic communities. A mobile team organized lung health camps in public facilities like schools and primary health facilities, and community meeting points. All individuals older than 15 years were verbally screened for TB symptoms (cough and duration, fever, night sweats, and weight loss) and offered a CXR screening using battery-powered, ultraportable X-ray (MinXray TR90BH, Northbrook, IL, USA) with AI-enabled interpretation using qXR v3 (Qure.ai Mumbai, India. AI products like qXR have been recommended by the WHO for interpretation of CXR images in TB screening and triage scenarios [2]. Sputum samples were requested for any individual with a qXR TB abnormality score of 0.30 or more (range 0.01-0.99) and/or reporting one or more TB symptoms. Sputum carriers transported samples to the nearest GeneXpert testing facility of the TB program and retrieved the results.

The intervention was delivered through existing health system structure as much as possible. The additional staff hired specifically for the intervention was the mobile lung health camp team composed of a registration officer, data entry staff, a radiographer, and a coordination officer. The Local Government TB and Leprosy Supervisor (LGTBLS) and community volunteers (CV) provided support in organizing the screening camp and received a stipend and incentive for additional work. Similarly, diagnostic testing using Xpert was conducted at existing laboratories, which received an additional support incentive for managing additional testing volume.

We used the health system perspective to conduct a retrospective costing of the ACF intervention delivered between July and December, 2022. We used a combination of top-down and bottom-up costing approaches while utilizing the itemized project expense data. We only estimated additional cost of implementing the ACF intervention on top of a functional TB program. Therefore, we did not include the cost of running a basic TB program (management, notification, treatment

initiation), cost of lab facilities (apart from cost of GeneXpert equipment, cartridges, and laboratory support incentives), and long-term operational costs like equipment maintenance.

## Cost parameters

We captured costs in three broad categories: training, screening, and testing. Training included one-time training of all personnel involved in the intervention at the start. The screening costs included human resources (HR), incentives for case finding, HR incentives, sputum transport, equipment and its transport between camps, community mobilization, and connectivity. The HR included cost of 1 LGTBLS per screening event and five CV for their engagement in the intervention and incentive for supporting each screening event, and lung health camp mobile team of four people. HR incentive included incentive to CV and LGTBLS for each presumptive individual identified for whom sample was collected. Sputum transport included cost of hiring motorbike and fuel. Community mobilization included cost of mobilization activities like public announcements and participation incentive to community members as a gift basket of household items. Connectivity was for internet data costs. Equipment was x-ray procurement and AI licence costs. The testing cost included cost of GeneXpert equipment and cartridge, sputum container, incentive to CV and LGTBLS for each confirmed TB diagnosis, and support incentive to participating laboratories. The cost for GeneXpert equipment was taken from the Global Drug Facility (GDF) product catalog for September, 2022 [27]. For Xpert, an additional surcharge of US$1.35 per cartridge was added on top of the GDF price, which was a universal cost per cartridge at the time to support machine maintenance and service. The remaining costs were actual incurred costs. The details of each cost parameter are presented in Table 1 and S1 File.

The quantity for presumptive identified and sample provided, sputum container, Xpert cartridge, and Xpert case detected were based on the algorithm modeled. The details for each line item are presented in S1 File.

## Analysis

The intervention was delivered through 66 lung health camp as punctuated events over a short timeframe of 142 days. Based on nature of costs, the costs were estimated either as proportion of time utilized (142 days out of 251 annual working days) or as unit cost per screening event or test units (number of samples, number of Xpert tests, number of confirmed TB diagnoses). Cost of the GeneXpert equipment was assigned as a proportion of annual test volume at full utilization (2 cycles a day on a 4-module machine for 251 working days per annum = 2008 tests per annum). We annuitized equipment cost and training cost at 3% discount rate with expected life years of use of 5 years and 2 years, respectively. All the costs were calculated in US$ (2022) with currency conversion rate of 470 Naira for a US$.

The outcome of ACF intervention was calculated using different screening criteria, i.e., CXR screening, symptom screening, and combination of CXR and symptom screening [26]. Similarly, we estimated costs in five screening criteria scenarios: abnormality score ≥ 0.30; abnormality score ≥ 0.50; cough ≥ 2 weeks; any symptom; abnormality score ≥ 0.30 *or* any symptom. For each scenario, we calculated total implementation cost of intervention and cost per case detected. Appropriate adjustments were made in calculating total costs, for example, the radiographer, CXR, and AI software costs were only included in CXR scenarios.

We also calculated incremental costs for algorithms and incremental effects as additional bacteriologically-confirmed TB diagnoses. We assessed cost-effectiveness in terms of natural units by calculating incremental cost-effectiveness ratio as additional cost per additional case detected.

Since we only included bacteriologically-confirmed cases, i.e., positive on Xpert, in the primary analysis, we modelled cost per case diagnosed by including clinically-diagnosed cases for scenarios with CXR as a secondary analysis. We used 19% estimated proportion of clinically-diagnosed cases out of all cases for Nigeria as per the WHO Global TB Report, 2023 to calculate number of clinical diagnoses [28]. Further, as a what-if analysis, we modelled cost per case detected for increasing proportions of asymptomatic TB, i.e., people with TB with no symptoms, for abnormality score ≥ 0.30 scenario by only increasing the total people diagnosed with asymptomatic TB at constant diagnostic yield.

**Table 1. Input cost parameters used for costing.**

| Intervention step | Cost category | Description | Unit Cost (US$) | Quantity | Time | Time Unit |
|---|---|---|---|---|---|---|
| Screening | HR | Community volunteers | 480 | 5 | 0.57 | year |
| Screening | HR | Local Government TB and Leprosy Supervisor (LGTBLS) | 1.9 | 1 | 66 | screening event |
| Screening | HR | Lung health camp team | 1080 | 4 | 0.57 | year |
| Screening | HR incentive | Screening day incentive | 10.6 | 5 | 66 | screening event |
| Screening | Incentives | Presumptive identified and sample provided | 2 | — | 1 | not applicable |
| Screening | Sputum and other transport | Motorbike hire | 45 | 1 | 66 | screening event |
| Screening | Sputum and other transport | Fuel for hired motorbikes | 20 | 2 | 66 | screening event |
| Screening | Transport | Fuel cost accessing screening sites# | 51 | 1 | 66 | screening event |
| Screening | Community mobilization | Mobilization activities | 34 | 1 | 66 | screening event |
| Screening | Community mobilization | Participation incentive to community members | 181 | 1 | 66 | screening event |
| Screening | Connectivity | Internet data | 1140 | 1 | 0.57 | year |
| Screening | Equipment | X-ray instrument - MinXray TR90BH | 75000 | 1 | 0.57 | year |
| Screening | Equipment | AI licence - qure.ai qXR | 9000 | 1 | 0.57 | year |
| Testing | Equipment | GeneXpert 4 module machine | 19500 | 1 | | proportion of annual test volume |
| Testing | Test | Sputum container | 0.62 | — | 1 | not applicable |
| Testing | Test | Xpert MTB/RIF cartridge | 11.33 | — | 1 | not applicable |
| Testing | Incentives | Case detected | 10 | — | 1 | not applicable |
| Testing | Incentives | Lab support incentive | 510.6 | 1 | 0.57 | year |
| Training | Training | Training for the team | 6287 | 1 | 1 | not applicable |

# The cost item 'fuel cost accessing screening sites' also includes the cost of charging the battery-powered x-ray and laptop devices using power from the vehicle.

Lastly, we conducted a sensitivity analysis by annuitizing equipment with 10 years of expected life years of use and also using costs of of different x-ray equipment and AI products.

Analysis was conducted in R v4.3.1. All the data is available in the manuscript and reproducible code is available on GitHub [29].

The study protocol for the intervention was reviewed and approved by Adamawa (no. 2022069) and Gombe State Research Ethical Committees in the Ministry of Health (no. GMEHREC/SMOH/2022/027). Verbal consent was provided by all participants as a large proportion of participants were illiterate. For those younger than 18, parental consent was given. The costing study utilized retrospective project expense data and did not require additional ethical approval. The study adhered to the tenets of the Declaration of Helsinki.

## Results

### Outcome of screening

We screened 5297 people using symptom screening questionnaires and CXR during the 66 screening events in the study period. Overall, we identified 3205 people with presumptive TB who either reported a symptom or had an AI abnormality

score of 0.30 or higher, out of which sputum sample was collected and sent for Xpert testing for 1021 individuals. Fewer sputum samples could be collected from people reporting symptoms (37% for people with only cough ≥ 2 weeks) compared to those only with abnormality score of 0.3 or higher (96%). Out of 1021 people tested with Xpert, 85 were positive for TB. (Table 2) The population characteristics and detailed case finding outcomes are available in the previous paper [26].

## Cost and cost-effectiveness of the intervention

The total cost was highest for the *Abnormality ≥ 0.30 OR any symptom* criteria at US$65704, followed by CXR-only criteria (Abnormality ≥ 0.30, US$61424; Abnormality ≥ 0.50, US$56589), and lowest for the cough ≥ 2 weeks criteria at US$40740. Conversely, the cost per case detected was highest for cough ≥ 2 weeks at US$1198, followed by the algorithms with CXR, and least for any symptom criteria at US$635. (Table 3) (Fig 1)

The main cost drivers of the intervention were screening cost (community mobilization in the form of participation incentive to people visiting the screening event, equipment cost for algorithm employing CXR including the AI licence) and testing cost (cost of the GeneXpert cartridge). The CXR and testing costs made up most of the overall cost difference between algorithms with and without CXR. (S2 File)

Compared to baseline strategy of cough ≥ 2 weeks, the incremental cost effectiveness ratio (ICER) for Any symptom was US$191 per additional case detected and US$ 2096 for *Abnormality ≥ 0.30 OR any symptom* algorithm. On the cost-effectiveness plane, Abnormality ≥ 0.50 and Abnormality ≥ 0.30 were not on the cost-effectiveness efficient frontier. Abnormality ≥ 0.50 algorithm was strictly dominated as it led to 1 fewer diagnosis but costed more than the *any symptom* algorithm. Abnormality ≥ 0.30 algorithm was removed by extended dominance because its ICER was US$ 3121 per additional case detected compared to US$1070 per additional case detected for *Abnormality ≥ 0.30 OR any symptom* after eliminating Abnormality ≥ 0.50. (Table 4) (Fig 2)

**Table 2. Outcome of screening for different screening algorithm.**

| Algorithm | Presumptive TB | Tests used | Bacteriologically-confirmed TB detected | People with TB missed |
|---|---|---|---|---|
| Abnormality ≥ 0.30 | 769 | 738 | 81 | 4 |
| Abnormality ≥ 0.50 | 447 | 424 | 76 | 9 |
| Cough ≥ 2 weeks | 1056 | 394 | 34 | 51 |
| Any symptom | 3083 | 906 | 77 | 8 |
| Abnormality ≥ 0.30 OR any symptom | 3205 | 1021 | 85 | — |

The tests used refers to Xpert MTB/RIF test. The number of people with TB missed is calculated with reference to *Abnormality ≥ 0.30 OR any symptom* algorithm based on bacteriologically-confirmed TB detected column.

**Table 3. Cost of implementing different screening algorithm.**

| Algorithm | Tests used | Bacteriologically-confirmed TB detected | Total Cost | Cost per case detected |
|---|---|---|---|---|
| Abnormality ≥ 0.30 OR any symptom | 1021 | 85 | 65704 | 773 |
| Abnormality ≥ 0.30 | 738 | 81 | 61424 | 758 |
| Abnormality ≥ 0.50 | 424 | 76 | 56589 | 745 |
| Any symptom | 906 | 77 | 48939 | 636 |
| Cough ≥ 2 weeks | 394 | 34 | 40740 | 1198 |

All costs are in US$. The tests used refers to Xpert test. The cost per case detected is calculated by dividing the total cost by the number of bacteriologically-confirmed TB detected, then rounding the result to zero decimal places.

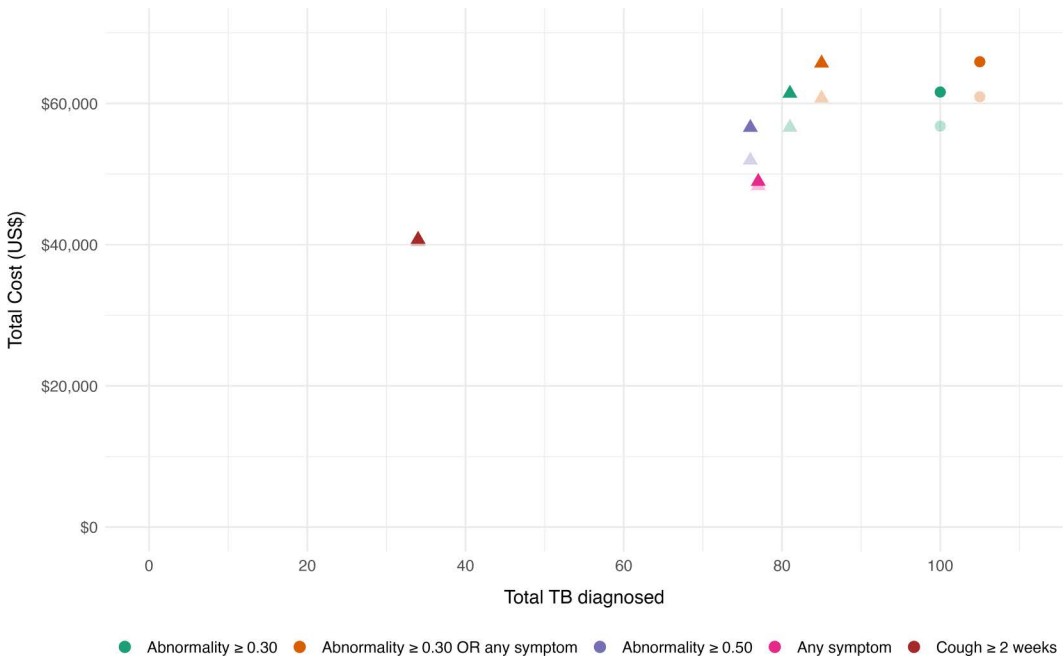

**Fig 1. Cost and cases diagnosed for each algorithm in different modelled scenarios.** The figure presents total cost and total TB cases detected for different algorithms as point estimates. The triangles represent scenarios that include only bacteriologically-confirmed TB, while the circles represent scenarios that also include clinically-diagnosed cases. Solid points (both triangles and circles) indicate scenarios with annuitization based on an expected life of 5 years, whereas transparent points use an expected life of 10 years.

## Modelled scenarios

In the modeled scenario assuming additional clinical diagnoses at 19% of overall TB diagnoses, the cost per case decreased for algorithms with CXR while maintaining the same relative position – in the range of US$604 – US$628 per case detected vs. US$745 – US$773 in the main analysis with only bacteriologically-confirmed diagnoses. Further, all the algorithms were on the cost-effective frontier on the cost-effectiveness plane with increasing ICERs as cost increased for higher number of TB diagnoses.(Figs 1, 3) (S4 File)

In the sensitivity analysis, the cost per case decreased by annuitizing equipment for 10 life years of use compared to 5 years in the main analysis. (Fig 1) This also changed the cost drivers with CXR costs decreasing from around 15% to around 9% for algorithms utilizing CXR. (S2 and S3 Files)

In simulating cost per case detected for increasing proportions of asymptomatic TB for Abnormality ≥ 0.30 algorithm, the cost per case declined with increasing proportion of asymptomatic TB and matched the cost per case detected for any symptom algorithm at around 30% asymptomatic TB. (Fig 4)

## Discussion

We provide implementation costing and cost-effectiveness estimates of symptom screening and CXR and AI screening in active case finding for TB amongst nomadic population in Nigeria. The combined screening algorithm using both symptoms and CXR-AI detected more bacteriologically-confirmed cases at US$773 per case detected, whereas symptom-only screening criteria missed 60% cases at a much higher US$1198 per case detected cost for cough ≥ 2 weeks criteria and 10% missed cases at cost lower than CXR algorithm of US$636 per case detected. In cost-effectiveness analysis for bacteriologically-confirmed TB, both CXR-only criteria (Abnormality ≥ 0.50 and Abnormality ≥ 0.30) were dominated,

PLOS Digital Health

**Table 4. Incremental cost-effectiveness analysis for TB screening algorithms.**

| Algorithm | Bacteriologically-confirmed TB detected | Total Cost | Cost per TB detected | Incremental Cost | Incremental Effect | ICER | Status |
|---|---|---|---|---|---|---|---|
| Cough ≥ 2 weeks | 34 | 40740 | 1198 | Ref | Ref | — | Non-dominated |
| Any symptom | 77 | 48939 | 636 | 8199 | 43 | 191 | Non-dominated |
| Abnormality ≥ 0.50 | 76 | 56589 | 745 | 7650 | -1 | — | Strictly Dominated |
| Abnormality ≥ 0.30 | 81 | 61424 | 758 | 4835 | 5 | — | Extended Dominated |
| Abnormality ≥ 0.30 OR any symptom | 85 | 65704 | 773 | 4280 | 4 | 2096 | Non-dominated |

The table compares five TB screening algorithms based on bacteriologically-confirmed TB detected, total cost, cost per TB case detected, incremental cost, incremental effect (additional TB cases diagnosed compared to the reference algorithm), and incremental cost-effectiveness ratio (ICER). The reference algorithm, cough ≥ 2 weeks, is used as a baseline for incremental comparisons. All costs are in US$. ICER is presented as additional US$ per additional TB case diagnosed.

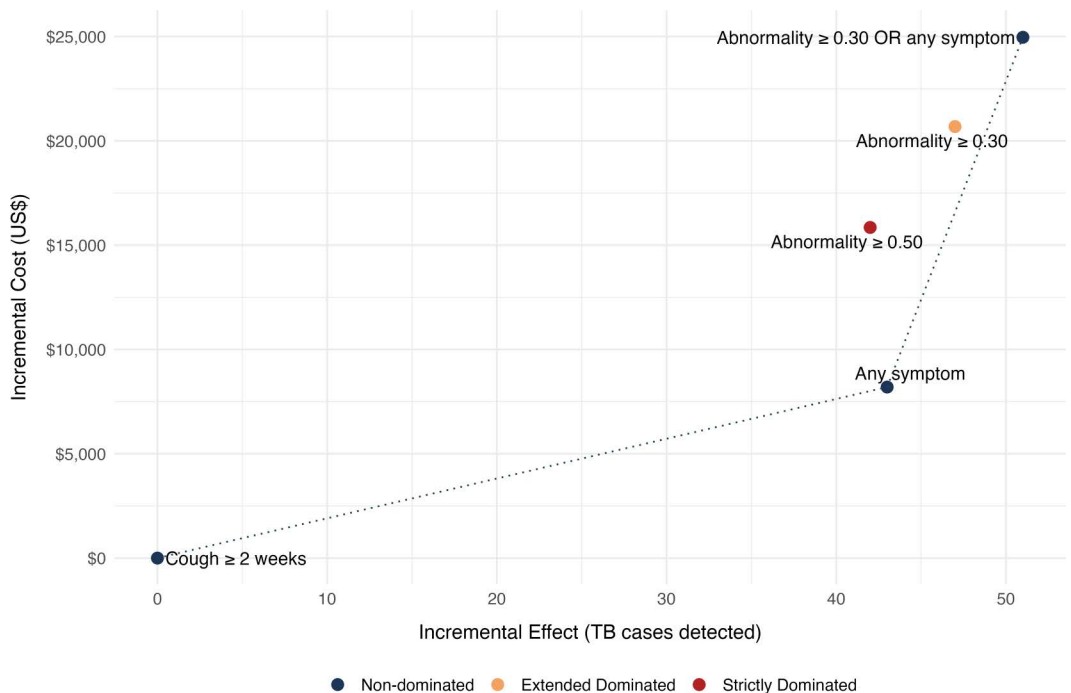

**Fig 2. Cost-effectiveness plane for TB screening algorithms.** The plot shows the incremental cost (US$) versus incremental effect (additional bacteriologically-confirmed TB cases detected) for various screening algorithms. Each point represents an algorithm. The dotted line connects non-dominated points, indicating the cost-effectiveness frontier.

whereas Abnormality ≥ 0.30 OR any symptom criteria, which identified all 85 cases had slightly higher yet comparable cost per case detected than symptom-only criteria.

The cost drivers of the intervention were CXR and AI costs and cost of the Xpert cartridges for testing, in addition to community mobilization cost, which are contextually specific to active case finding in nomadic population in this region of Nigeria. Our intervention did not require a radiologist or physician to be present, but other country regulation have different requirements for personal which can affect the HR costs [30]. Since the human resource and other operational

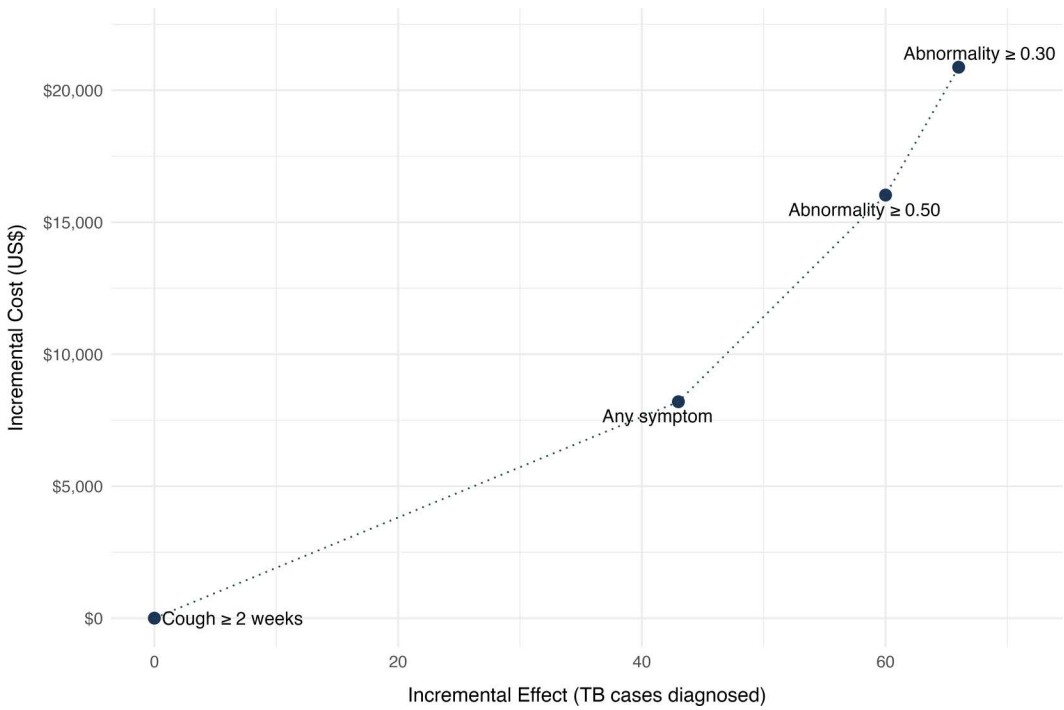

**Fig 3. Cost-effectiveness plane for TB screening algorithms including clinically-diagnosed cases.** The plot shows the incremental cost (US$) versus incremental effect (additional TB cases diagnosed) for various screening algorithms when including clinically-diagnosed cases in addition to the bacteriologically-confirmed cases. Each point represents an algorithm. The dotted line connects non-dominated points, indicating the cost-effectiveness frontier.

costs were similar and proportionately smaller, the cost tradeoff in programmatic implementation is potential cost savings in testing against procurement of expensive ultraportable CXR devices with AI which can cost in the range of US$ 60,000 – US$100,000 [31]. Nonetheless, CXRs are unlike molecular tests in the sense that unit cost per CXR exposure will decrease as utilization increases. Therefore, cost per person of CXR and AI license can be considerably smaller when utilized to its full value over longer durations, for instance, in active case finding campaigns in community-settings [32]. Conversely, insufficient utilization of ultraportable CXR will hamstring its impact with overall higher costs for the outcome, at which point it would be worthwhile investing in alternate interventions. In many rural settings, particularly with limited CXR access [33], offering a portable CXR is an externality that can bring more people to ACF events. CXR is often very costly even when people can access it [34], so a free CXR is likely an incentive by itself.

Several other use cases and approaches will have an impact on efficiency and cost-effectiveness. CXRs can be utilized for broader lung health beyond just TB where AI-enabled interpretation can be potentially useful [35]. Multiplexing CXR and AI for use cases beyond TB, particularly at primary and secondary care level where pulmonologists are scarce, can further improve cost-effectiveness and drive down unit costs. In TB interventions, CXR-AI can also be used a triage for pooling specimens for diagnostic testing on molecular platforms to gain efficiency [36].

In our modelled scenario, the cost per case diagnosed and cost-effectiveness improves on including clinically-diagnosed TB cases and increasing proportion of asymptomatic TB, both of which are an advantage of including CXR in algorithms compared to just symptom screening. In our analysis, there was about 10% asymptomatic TB for the Abnormality ≥ 0.30 criteria. This is in contrast with the Nigeria TB prevalence survey where 36% of people with TB reported no symptoms [4]. The difference we observed may be because of the target population being sicker given their limited access to care and possibly more advanced disease. Our analysis showed that using CXR and AI had lower cost per

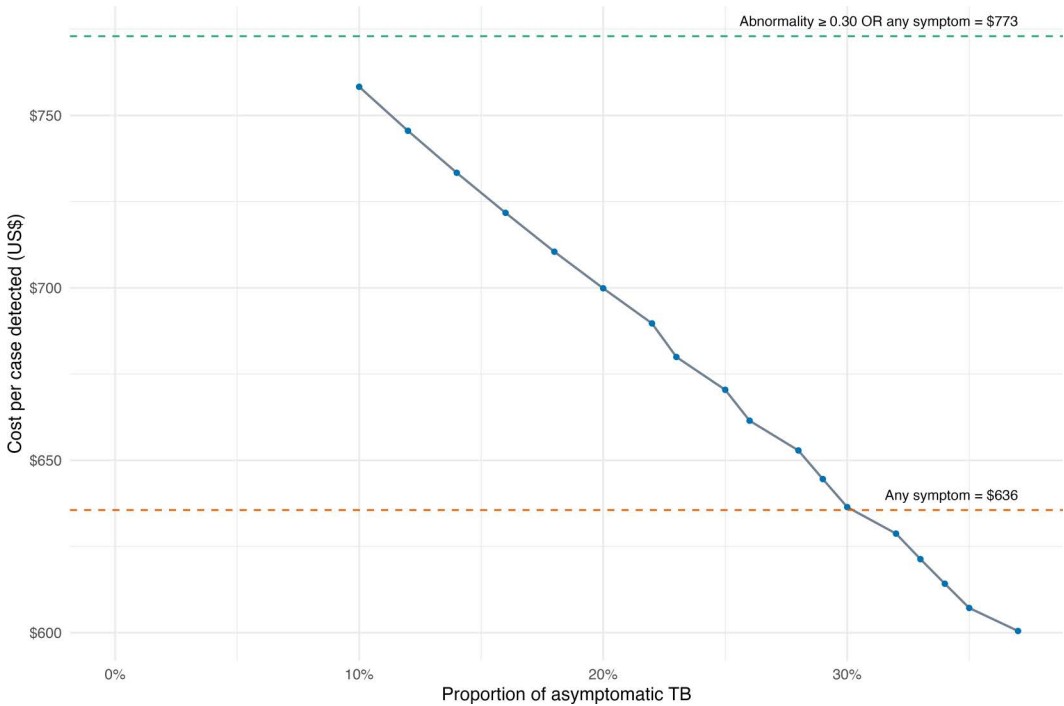

**Fig 4. Modelled cost per TB Case Detected for the Abnormality ≥ 0.30 algorithm.** The plot illustrates the modelled cost per TB case detected for bacteriologically-confirmed TB across various scenarios with increasing proportion of asymptomatic TB within the main analysis. The cost for Abnormality ≥ 0.30 algorithm in empirical analysis is US$ 758 per case diagnosed at 10% asymptomatic TB, which reduces with increasing proportions of asymptomatic TB to match that of any symptom algorithm at 30% asymptomatic TB.

case detected than any symptom screening criteria when asymptomatic TB was higher than 30% of all bacteriologically-confirmed TB detected. (Fig 4) While the exact threshold will vary based on test costs, generally, CXR and AI will be a more cost efficient strategy in screening settings with higher proportions of asymptomatic TB. Additional studies in different populations should conduct additional CEA for CXR and ACF.

The TB diagnostics space is rapidly changing with lower prices for molecular tests being negotiated [31,37], lower cost CXR equipment being available [38], and newer diagnostic tests being developed. All of these advancements could impact the cost calculations in different ACF approaches as sample collection rates could improve with swab-based tests [39], and lower costs of X-ray and AI may materialize with more competition in the market [40].

In our intervention, a large proportion of people with symptoms but clear CXR could not produce a quality sputum sample, hence could not be offered a molecular test. While this is an implementation limitation, this results in more optimistic costs for symptom screening criteria because it would have resulted in higher costs if those individuals could be offered testing. The cost per case estimates, bacteriologically-confirmed or all forms, are from a health systems perspective and studies have shown that ACF can save out of pocket costs for people with TB, meaning that total costs are probably overestimated [18]. We used a limited health system perspective and costed intervention over and above a basic functional TB program. However, our costs are pragmatic, and we included additional cost of intervention, like CXR and AI cost and Xpert equipment and cartridge cost to calculate realistic and robust estimates. Our study was in a resource-limited setting with a unique population group which limits the generalizability to other contexts. ACF implementation is highly contextualized and needs customization compared to similar intervention at facility [17]. ACF is also often more expensive because of the need for extra effort in providing services to people with poorer access to health system. In replicating such an

intervention in another setting like urban area or facility, TB prevalence, including that of asymptomatic TB, and service utilization will be important determining factors for cost efficiency. Future studies should undertake prospective costing using micro-costing approaches over longer intervention duration in different population and country settings while considering both health system and societal perspectives.

## Conclusion

Although a single study, this is the first cost-effectiveness analysis comparing CXR with AI and symptom screening for TB detection. Compared to traditional symptom screening, using CXR and AI or combination with symptoms detects more cases at lower cost per case detected. CXR and AI in ACF can be a cost-effective intervention at current cost levels with further scope for improving cost-effectiveness especially when considering the possibility of clinical diagnosis. While more studies are needed in different populations and using different approaches, National TB Programs should explore adoption of CXR and AI in ACF for screening.

## Supporting information

**S1 File. Details of input cost parameters.** The *cost* sheet contains input cost parameters used for costing. The *cost detail* sheet contains further details for calculations related to individual line item.
(XLSX)

**S2 File. Detailed cost outputs for the main analysis.** These are detailed cost outputs for main analysis where only bacteriologically-confirmed diagnoses are included, and annuitization used 5 years of expected life years of use. The cost sheet has absolute costs for each algorithm and line items. The cost drivers are in the cost line item sheet.
(XLSX)

**S3 File. Detailed cost outputs for the sensitivity analysis.** These are detailed cost outputs for sensitivity analysis where only bacteriologically-confirmed diagnoses are included, and annuitization used 10 years of expected life years of use. The cost sheet has absolute costs for each algorithm and line items. The cost drivers are in the cost line item sheet.
(XLSX)

**S4 File. Detailed cost outputs for the modelled analysis.** These are detailed cost outputs and ICER for sensitivity analysis where both bacteriologically-confirmed and clinically-confirmed diagnosis are included, and annuitization used 5 years of expected life years of use. The cost sheet has absolute costs for each algorithm and line items. The cost drivers are in the cost line item sheet.
(XLSX)

**S5 File. Cost per case detected across different x-ray and AI equipment scenarios.** These are results of sensitivity analysis using cost data from the Global Drug Facility's Diagnostics, Medical Devices & Other Health Products Catalog, March 2025. All scenarios include prices from this catalog for Xpert Ultra cartridge, X-ray and AI equipment. The base case uses actual prices for X-ray and AI equipment with current prices for Xpert Ultra cartridge. Ultra has replaced MTB/RIF, which is no longer available in the catalog.
(PDF)

## Acknowledgments

The authors would like to acknowledge the support of all staff at the participating facilities and also the participants who engaged in this project. We want to recognize and thank the Adamawa and Gombe state governments for their support and the numerous community volunteers for their tireless work in identifying people with TB and supporting them in Adamawa and Gombe states.

## Author contributions

**Conceptualization:** Tushar Garg, Beatrice Kirubi, Jacob Creswell.

**Data curation:** Tushar Garg, Stephen John, Suraj Abdulkarim, Adamu D. Ahmed, Beatrice Kirubi, Md. Toufiq Rahman.

**Formal analysis:** Tushar Garg.

**Funding acquisition:** Stephen John, Suraj Abdulkarim.

**Supervision:** Jacob Creswell.

**Writing – original draft:** Tushar Garg, Jacob Creswell.

**Writing – review & editing:** Tushar Garg, Stephen John, Suraj Abdulkarim, Adamu D. Ahmed, Beatrice Kirubi, Md. Toufiq Rahman, Emperor Ubochioma, Jacob Creswell.

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
