## [Decision Letter · Decision Letter 0]

Response to Reviewers
Revised Manuscript with Track Changes
Manuscript
**Additional Editor Comments (if provided):**

1) The willingness to pay threshold based on GPD pre capita used here is usually employed with respect of one QALY gain. In your setting you have additional detected cases as the outcome. Please link the gain to QALY gains from the additional detection (and then use this in combination with the GDP) to get to a threshold or use another measure for social willingness to pay for one additional detected case.

2) The license costs you use in your estimation differ widely from the ones reported in the discussion (based on reference 32). Please add a cost-effectiveness analysis taking into account the possible real costs for full implementation.

Minor Points:

- I agree with reviewer 2 that electricity sounds like costs that should be in there.

**Reviewers' Comments:**

**Comments to the Author**

1. Does this manuscript meet PLOS Digital Health’s publication criteria?

Reviewer #1: Yes

Reviewer #2: Yes

2. Has the statistical analysis been performed appropriately and rigorously?

Reviewer #1: I don't know

Reviewer #2: Yes

3. Have the authors made all data underlying the findings in their manuscript fully available (please refer to the Data Availability Statement at the start of the manuscript PDF file)?

Reviewer #1: Yes

Reviewer #2: Yes

4. Is the manuscript presented in an intelligible fashion and written in standard English?

Reviewer #1: Yes

Reviewer #2: Yes

Reviewer #1: A worthwhile and timely article. I acknowledge the time and effort the authors have invested in this research.

While the authors employed a health system perspective for the analysis, a societal perspective would have been preferable to capture all externalities, including those associated with environmental costs of implementing the program.

Reviewer #2: This study discusses the cost-effectiveness and implementation costs of using ultraportable chest X-rays (CXR) combined with artificial intelligence (AI) in active case finding (ACF) for tuberculosis (TB) in Northeast Nigeria. The study involved people 15 years and older in Northeast Nigeria. The authors compared five screening scenarios, including symptom-based methods and CXR-AI approaches, to identify the most efficient strategy for detecting TB cases. The findings demonstrated that combining CXR and AI with symptom screening detects more cases at a lower cost per case diagnosed.

Overall, the paper contributes to understanding the cost implications and effectiveness of adopting CXR and AI for TB screening in ACF.

The paper is highly relevant as it addresses a critical global health issue: improving TB detection through the adoption of CXR and AI. The authors have also clearly explained the methodology and provided the dataset and code, which aids the reproducibility of the findings.

While this study has numerous strengths, there are a few concerns: The study is more focused on resource-limited regions and is limited in generalizability to other areas due to the peculiarity of the studied region. Expanding the discussion on how these findings could be adapted to different populations or healthcare settings would be beneficial.

I didn't see the cost of electricity in the cost parameter. Was electricity not a factor in the experiment? If it was, the authors should clarify how it was accounted for.

The authors discussed limitations briefly but could provide more detail on the challenges of replicating these findings in urban or higher-resource settings.

Also, there are minor grammatical errors, such as "promising developments tuberculosis (TB) active case finding" (which should be "promising developments in tuberculosis (TB) active case finding").

**Do you want your identity to be public for this peer review?** For information about this choice, including consent withdrawal, please see our Privacy Policy

Reviewer #1: No

Reviewer #2: No

**Figure resubmission:****Reproducibility:** To enhance the reproducibility of your results, we recommend that authors of applicable studies deposit laboratory protocols in protocols.io, where a protocol can be assigned its own identifier (DOI) such that it can be cited independently in the future. Additionally, PLOS ONE offers an option to publish peer-reviewed clinical study protocols. Read more information on sharing protocols at https://plos.org/protocols?utm_medium=editorial-email&utm_source=authorletters&utm_campaign=protocols

---

## [Editor Report · Decision Letter 1]

Response to Reviewers
Revised Manuscript with Track Changes
Manuscript
**Additional Editor Comments (if provided):**

**Reviewers' Comments:****Figure resubmission:****Reproducibility:**To enhance the reproducibility of your results, we recommend that authors of applicable studies deposit laboratory protocols in protocols.io, where a protocol can be assigned its own identifier (DOI) such that it can be cited independently in the future. Additionally, PLOS ONE offers an option to publish peer-reviewed clinical study protocols. Read more information on sharing protocols at https://plos.org/protocols?utm_medium=editorial-email&utm_source=authorletters&utm_campaign=protocols

---

## [Editor Report · Decision Letter 2]

Implementation costs and cost-effectiveness of ultraportable chest X-ray with artificial intelligence in active case finding for tuberculosis in Nigeria

PDIG-D-24-00504R2

Dear Dr Creswell,

We are pleased to inform you that your manuscript 'Implementation costs and cost-effectiveness of ultraportable chest X-ray with artificial intelligence in active case finding for tuberculosis in Nigeria' has been provisionally accepted for publication in PLOS Digital Health.

Best regards,

Simon Reif

Academic Editor

PLOS Digital Health

**Additional Editor Comments (if provided):**

Thanks for the quick revision!